# Midwives' and Public Health Nurses' Experiences of Implementing a Guided Version of a Digital Intervention, Mamma Mia, in Maternity and Child Health Care Services: A Reflexive Thematic Analysis

Ellen Solstad Olavesen[1,2]*, Sølvi Helseth[2,3], Silje Marie Haga[1], Thea Sundrehagen[1,4], Filip Drozd[1]

1 Section for Infants and Young Children, Regional Centre for Child and Adolescent Mental Health, Eastern and Southern Norway, Oslo, Norway, 2 Faculty of Health Sciences, Oslo Metropolitan University, Oslo, Norway, 3 Department of Psychosocial Health, University of Agder, Kristiansand, Norway, 4 Department of Psychology, University of Oslo, Oslo, Norway

* eso@r-bup.no

## Abstract

Guided digital interventions in maternity and child health care may improve mental health outcomes by increasing user engagement and adherence. However, implementing such complex interventions remains challenging due to factors related to intervention characteristics, organizational dynamics, professionals' ompetencies, and user engagement. Understanding the experiences of those who deliver these interventions is essential to inform implementation efforts. This study explored midwives' and public health nurses' experiences of a guided digital intervention and the early implementation process three months post-training. Using a qualitative design, six focus groups were conducted with 32 professionals working in maternity and child health care services and participating in a multisite cluster-randomized trial of the universal digital intervention, *Mamma Mia*. Interviews were based on the Consolidated Framework for Implementation Research, and data were analyzed using reflexive thematic analysis. Three themes described how professionals made sense of implementing a guided digital intervention. *"Broken Expectations – Navigating the Training-Practice Gap"* reflected how abstract, decontextualized training left professionals' feeling unprepared once faced with real-world implementation. *"Balancing Belief and Doubt – Navigating digital care in relational professions"* captured tensions between valuing innovation and safeguarding professional values, as participants negotiated whether digital tools could align with relational care practices. *"Learning by Doing and Support – Future Optimism"* illustrated a gradual shift toward acceptance, as hands-on experience and collegial support helped professionals view the intervention as complementary rather than disruptive. These findings underscore that implementing digital interventions in relational care settings is not solely a technical process but

**Data availability statement:** The data that support the findings of this study are available in the manuscript, supporting files, and from Regional Centre for Child and Adolescent Mental Health, Eastern and Southern Norway, Oslo, Norway, but restrictions apply to the availability of these data, which were used under license for the current study and so are not publicly available. Data are however available from the author upon reasonable request and with permission of the Regional Centre for Child and Adolescent Mental Health, Eastern and Southern Norway. Contact information: https://www.rbup.no/om/ansatte/Ole-Martin-Vangen.

**Funding:** The research project received grants from the Norwegian Research Council (Ref. 295944), which are gratefully acknowledged. TS received a salary from the Norwegian Research Council (Ref. 295944). The funders had no role in study design, data collection and analysis, decision to publish, or preparation of the manuscript.

**Competing interests:** The authors have declared that no competing interests exist.

a relational and interpretive one. Effective implementation should therefore support contextual adaptation, provide practical and collaborative training, foster trust, and ensure organisational alignment.

## Author summary

This study explored how midwives' and public health nurses' experiences of implementing *Mamma Mia*, a guided digital intervention designed to support perinatal health, in Norwegian maternity and child health care services. Thirty-two professionals participated in focus group discussions three months after completing training as part of the MaMi Trial II. The findings showed that early experiences with the guided intervention were shaped by a gap between the professional expectations of training and the realities of using the program in everyday practice. Participants described uncertainty about how the digital tool fit with their established ways of working and their role in providing relational care. Over time, however, hands-on experience, support from colleagues, and ongoing supervision helped professionals develop greater confidence and see the intervention as a helpful addition rather than a replacement for face-to-face care. Overall, the study highlights that implementing digital tools into maternity and child health care is not only about learning new skills but also about how professionals make sense of change in relation to their values, responsibilities, and clinical relationships. Practical and collaborative training, opportunities to learn by guided experience, and organizational support are essential to help foster trust and integration of digital interventions into routine care.

## Introduction

Depressive symptoms during the perinatal period remain a significant public health challenge, despite established guidelines and preventive efforts (Carlson et al., 2025). Maternity and child health services (MCHCs) are uniquely positioned to address perinatal mental health during a period of frequent contact and established trust with women [1]. In response to growing service demands, digital health interventions have been promoted as a way to strengthen mental health within these settings [2,3].

Digital health interventions, including the *Mamma Mia* app, offer accessible and scalable support by delivering evidence-based content alongside routine care [2–5]. When combined with professional guidance, blended care approaches have shown potential to improve outcomes and engagement, particularly for individuals with more severe symptoms [6–9]. For MCHCs, such interventions are increasingly seen as means to complement face-to-face care, enhance continuity of support, and improve access for women who may be reluctant to seek help through traditional services [8,10].

However, translating the promise of digital interventions into routine clinical practice remains challenging. Implementation is a complex process shaped not only by characteristics of the intervention, but also by organizational conditions, professional roles, and everyday clinical realities [11–14]. Systemic constraints, competing priorities, ambivalence toward digital interventions among health personnel, and concerns about replacing traditional care can hinder adoption, particularly in perinatal care where professionals must balance medical, psychological, and relational needs in a constrained system [3,11,15,16].

Although blended interventions can improve outcomes, many innovations remain in the pilot stage, and fail to become embedded in routine practice and benefit users [3,9,15]. Healthcare professionals play a pivotal role in determining whether such interventions are adopted, adapted, or abandoned, yet their experiences of implementing guided digital interventions are often underexplored [17]. Professionals are frequently expected to promote and deliver these tools without sufficient preparation, training, or organizational backing, which may undermine confidence, engagement, and sustainability [11,15]. Mamma Mia was originally developed as a self-guided digital intervention and is now being tested in a blended format within MCHCs, where midwives and public health nurses (PHNs) guide women in its use during pregnancy and the postpartum period. This shift places new demands on professionals and requires them to integrate guidance in the intervention into established routine practice.

Therefore, this study aimed to explore how midwives and PHNs experience the early implementation of the guided Mamma Mia in MCHCs. By examining their perceptions, challenges, and sensemaking during the initial phase of implementation, this study seeks to generate insights that can inform future implementation efforts and support that the meaningful integration of digital interventions into everyday perinatal care.

## Context

In Norway, there are 357 municipalities. Each municipality is responsible for health-promoting and preventive services in the antenatal period and for children and their families [18]. Together with general practitioners (GPs), PHNs and midwives are responsible for providing primary care during the perinatal period. All pregnant women are entitled to free follow-up consultations with their midwife and/or GP. During pregnancy, women are offered nine consultations, including foetal diagnostics and ultrasounds, with additional needs-based follow-up and a home visit in the early postpartum period [19]. After birth, women and children are provided the Infant healthcare program, a standardized program consisting of 14 individual and group consultations, including a home visit, delivered mainly by PHNs [20].

## Aim

The study aimed to explore midwives' and public health nurses' experiences of the intervention and the early implementation process, three months post-training in a guided digital intervention.

## Materials and methods

### The guided intervention

*Mamma Mia* was developed as a fully automated digital intervention designed to prevent perinatal depression and enhance subjective well-being among perinatal women. It consists of 44 structured sessions delivered over approximately 11 ½ months, starting in the second trimester and continuing until the baby is six months old. The program targets risk and protective factors, including attachment, couple satisfaction, social support, and subjective well-being, utilizing psychoeducation, mindfulness, and meta-cognitive strategies. Mamma Mia is free of charge, providing a scalable and accessible tool for universal prevention of perinatal depression [21].

The goal of providing *Mamma Mia* with guidance from midwives and PHNs at MCHCs is twofold: 1) support the use of the intervention and 2) facilitate and promote perinatal mental health and subjective well-being by providing

person-centred and tailored care based on the individual woman's needs. The exploration and support use the Efficiency Model of Support (EMS model), which aims to enhance digital intervention use and clinical efficacy by addressing five key failure points: usability, engagement, fit, knowledge, and implementation [22]. The blended version aims to identify and address these failure points stepwise, ensuring users benefit fully from the intervention and explore additional needs. Midwives and PHNs received training in this approach during the two-day training as described in the training manual [23,24].

## Training and supervision

To facilitate the implementation of the guided version of *Mamma Mia*, a comprehensive plan and a detailed guide for recruitment and the guided intervention were developed to support the study and its implementation [24]. Training and supervision were delivered by three trainers through a two-day digital training program (Table 1), followed by monthly 1-hour digital supervision, to ensure adherence to the intervention protocol and provide ongoing support for healthcare professionals during implementation. The three trainers who delivered the training sessions and supervision were selected based on their professional background as midwives and PHNs and their experience with quality improvement in MCHCs. Manuals for training and supervision were developed to support trainers.

## Study design

This study employed a qualitative descriptive design to explore midwives' and PHNs' experiences of implementing a blended digital intervention, *Mamma Mia*, into routine maternity and child health. A qualitative, reflexive thematic analysis approach was chosen to capture participants' own accounts of meaning, uncertainty, and adaptation during early implementation, rather than to test predefined hypotheses.

Data were analyzed using reflexive thematic analysis (RTA), as described by Braun and Clarke [25], which was selected for its flexibility and suitability for examining how professionals make sense of complex, context-dependent change. RTA enabled an inductive, interpretive engagement with the data, allowing patterns of meaning to be developed through iterative reflection rather than through quantification or saturation-based logic [26, 25].

The study adhered to the COREQ (Consolidated Criteria for Reporting Qualitative Research) checklist for transparency and rigor (Tong, Sainsbury et al., 2007), followed the principles of the Declaration of Helsinki [27], and received ethical approval from the Regional Ethical Committee (ref. no. 112830). The study forms part of a larger multisite randomized

**Table 1. The structure of the two-day training sessions.**

- Welcome, presentation of participants and supervisors, introduction and outline
- Plenary discussion of participants' experiences from the self-study of Mamma Mia
- Background and evidence-based development of Mamma Mia
- Plenary sessions about the resources available to clinicians and how to access them
- Plenary introduction and discussion of recruitment and research participation
- Plenary introduction to the guided Mamma Mia, roleplay and small group discussions about the EMS model, integrating Mamma Mia in consultations and providing tailored counselling using standardized cases
- Summary and introduction next session

- Introduction to implementation
- How to implement Mamma Mia
- Why is implementation important?
- Dissemination vs. implementation
- Introduction implementation plan
- Group discussion: Local adaptation, assessment of factors influencing implementation
- Group discussions identifying facilitators and barriers to implementation in each municipality
- How to use the implementation plan to support implementation
- Group discussion: Expectations of implementation of Mamma Mia
- Supervision: Aim, group-based, implementation support

controlled trial (MaMi Trial II, ISRCTN ref: 29971). All participants provided written informed consent at baseline and orally consented before the FGs.

## Participants and recruitment

This study included 32 healthcare professionals from eight municipalities spread across 15 MCHCs in Norway. Midwives and PHNs were purposively selected to ensure maximum variation based on geographical location (urban vs. rural), clinic size, and training group affiliation. This sampling strategy aimed to capture a broad range of perspectives across different implementation contexts. In clinics with fewer than 14 employees, all eligible staff were invited to participate; in larger clinics, a subset was selected to ensure feasibility while maintaining diversity. To support open dialogue and interaction, focus groups were kept deliberately small, promoting equal opportunity for all to contribute. Invitations were distributed via Microsoft Teams, followed by SMS reminders sent 48 hours prior to the scheduled session.

## Data collection

Demographic data were collected via an electronic questionnaire. Qualitative data were collected between May 2022 and October 2023 through six focus groups ($n = 4$–$8$) lasting from 1 hour and 11 minutes to 1 hour and 33 minutes. Focus groups were chosen to explore how professionals jointly negotiated understandings of the intervention, implementation challenges, and their professional role in guided digital care. Group interaction enabled participants to build on one another's accounts, challenge assumptions, and reflect on shared organizational practices, thereby enriching the analytical material.

## Interview procedure and Use of the CFIR

The focus groups followed a semi-structured interview guide informed by selected constructs from the Consolidated Framework for Implementation Research (CFIR), particularly those related to individual competence, training, organizational fit and support, and early implementation experience (https://cfirguide.org/tools/tools-and-templates/; Table 2) [11]. CFIR was used as a sensitizing framework to ensure that interviews addressed known implementation domains, while still allowing participants to raise issues they perceived as most salient. Importantly, CFIR did not function as an analytic coding framework. Instead, it guided the focus group discussions and supported comprehensive data generation without constraining the inductive analytic process. This approach allowed themes to be developed reflexively from participants' accounts, while remaining attentive to implementation-relevant dimensions such as readiness, support, and contextual fit. Follow-up questions were used flexibly to encourage elaboration and clarify meanings. Information about the research aim, background, and research interest of the moderators was provided before each interview.

**Table 2. Main interview questions based on CFIR.**

| How has the training and supervision prepared you for implementing Mamma Mia? | Individual characteristics – Access to knowledge and information and self-efficacy/feeling) |
| --- | --- |
| How does Mamma Mia compare to similar programs in your setting? | Intervention Characteristics – Relative advantages |
| What are the pros and cons of Mamma Mia compared to existing interventions? | Intervention Characteristics – Relative advantages |
| What structural changes have been necessary to adopt Mamma Mia? | Inner setting – Structural characteristics |
| How receptive is your organization to implementing this intervention? | Inner setting – Implementation Climate |
| In what ways has the management supported the implementation of Mamma Mia so far? | Inner setting – Readiness for implementation |

All interviews were conducted in Norwegian. Prior to the interviews, the FG moderators had digitally met with participants on two occasions, one meeting which was an invitation to participate in the study, and an information meeting after being randomised to provide the guided version of *Mamma Mia*. The first author (ESO) facilitated all focus groups, supported by a second moderator (TS). After each session, the second moderator summarized key points raised and asked participants to confirm, nuance, or expand on these interpretations. These processes supported immediate reflexive engagement with the data rather than formal member checking. Discussions were video- and audio-recorded and transcribed verbatim. Field notes and transcripts were reviewed iteratively to support familiarization and reflexive awareness of emerging interpretations. Reflexivity was treated as an ongoing analytic practice, shaping how patterns were identified, questioned, and refined across the analytic process, rather than as a discrete procedural step. All data were securely stored in accordance with ethical guidelines.

### Data analysis

The demographic variables (age, gender, years of professional experience, and professional role) were analyzed using SPSS, Version 27.0 [28], and are presented as descriptive statistics (frequencies, percentages, range, and means). Qualitative data were analysed using reflexive thematic analysis, following Braun and Clarke's six-phase approach: familiarization, coding, theme development, theme review, theme definition and naming, and writing up [25]. Co-authors FD, SH, and SMH contributed to the review and development of themes. Themes were constructed inductively through engagement with the dataset, rather than pre-defined categories. QualCoder 3.5 [29] was used to support data management and coding. Its open-source and transparent structure enabled systematic documentation of coding decisions and analytic memos, which was particularly important as a first-cycle coding was conducted by a master student and second-cycle coding and initial theme development were undertaken by the first author [30]. To ensure transparency, examples of codes and theme development are included in S1 File.

### Rigour and trustworthiness

Trustworthiness was enhanced through investigator triangulation and reflexive dialogue. The research team consisted of individuals with diverse professional backgrounds and genders, PHNs (ESO, SH), a psychologist and MSc (TS), and experienced qualitative researchers with PhDs (SH, FD, SMH). ESO, TS, SMH and SH are female, while FD is male, and FD, ESO and TS, ESO conducted the interviews and initial analysis. The interpretation of the findings was strengthened through collaborative discussions exploring multiple perspectives and assumptions. Memos were created during and after each interview to capture reflections and reduce bias, and these were regularly discussed within the research team. The team met frequently to review emerging themes and findings, supporting a transparent analytic process. The interviewer also kept a reflexive diary throughout the study to document personal reflections, expectations, and potential biases, including how her professional background, involvement in developing the training material, and beliefs about implementing the intervention in MCHCs might have influenced interactions with participants. Rather than seeking consensus, the team aimed to co-construct a rich and credible interpretation of the data.

## Results

### Demographics

The six focus groups included 15 midwives, 14 public health nurses, and 3 nurses in public health nursing education. Participants from the same MCHC were interviewed together; however, due to limited staff numbers, two focus groups were conducted with participants from two different MCHCs. They were all female, with a mean age of 46 years (range 27–62 years). The mean years of experience in the current position of MCHCs was 6.8 years (range 0–23).

## Themes

Three themes were developed, which together shed light on how PHNs and midwives experienced the initial implementation of the guided *Mamma Mia* intervention in clinical practice. Three overarching themes were developed through reflexive thematic analysis: "*Broken Expectations – Navigating the Training-Practice Gap*", "*Balancing Belief and Doubt – Navigating Digital Care in Relational Professions*", and "*Future Optimism – Learning by Doing and Support*. Together, these themes highlight the dynamic nature of implementation, in which initial uncertainty and unmet expectations gradually gave way to increased ownership and belief in the intervention's value, driven by experiential learning and supportive structures.

### Theme 1: Broken Expectation – Navigating the Training-Practice Gap

Participants described initial enthusiasm and curiosity when introduced to *Mamma Mia*, but this optimism was quickly challenged during training and early implementation. A central source of frustration was a perceived mismatch between expectations of the training and its practical relevance. Participants had anticipated concrete, practice-oriented instruction on how to introduce, guide, and integrate the intervention within consultations. Instead, many experienced the training as abstract, overly complex, and insufficiently tailored to their clinical context. As one PHN explained: "*I felt like those training days were somewhat unstructured... I spent two days without really understanding what we were doing. I think we had high expectations, but in the end, we were a bit disappointed.*" (PHN, focus group 6). This disconnect undermined>participants' sense of preparedness and shaped their early orientation toward the intervention. While the training covered technical and conceptual aspects of Mamma Mia, participants struggled to translate this knowledge into concrete clinical action. Several described uncertainty about how guidance was meant to look in practice, particularly when trying to balance structured techniques with their usual conversational style. As one midwife put it: "*I was left thinking: am I really not doing anything of what I'm supposed to do? Because I don't know what I'm doing. And then I have to choose between which four techniques… and I really feel like I'm just talking*" (Midwife, focus group 4).

These accounts illustrate how the training-practice gap affected not only skill acquisition but also professional confidence. Participants felt responsible for delivering an intervention they did not yet feel competent to deliver, creating tension between professional expectations and perceived capability. This uncertainty, in turn, contributed to frustration, reduced motivation, and a hesitancy in promoting the intervention to women. Supervision, which participants had expected to compensate for these challenges, was often misaligned with their needs at this early stage. Several described supervision sessions as focused on implementation logistics rather than on developing conversational skills. One midwife said: "*I expected supervision to go beyond just implementation—more like training us to be skilled conversation partners and effective support for women, not just focusing on how to apply the techniques. I was hoping for something more hands-on*" (Midwife, focus group 6).

Timing further compounded these challenges. For many PHNs, primarily involved in postnatal care, there was a substantial delay between training and the opportunity to guide the use of Mamma Mia. This temporal gap weakened recall, reduced confidence, and limited opportunities to practice skills. As one PHN put it: "*We have not started yet; I think that I agree with what was said earlier… that the training and supervision came a little early for us because we do not get very much out of it right now*" (PHN, focus group 1). Midwives described related challenges among certain groups. Contextual and cultural factors also shaped implementation experiences. A midwife working with the Sámi population highlighted how local norms influenced engagement: "*There's a culture here of managing on your own… That might also be why recruitment is challenging; mental health remains a difficult topic for many.*" (Midwife, focus group 3).

Taken together, these experiences illuminate how early misalignment between expectations and training content, and clinical realities weakened initial readiness for implementation and delivery of the guided Mamma Mia. This training-practice gap functioned not merely as a practical obstacle, but as an early interpretive lens through which participants

evaluated the intervention's feasibility and relevance. While this gap generated frustration and uncertainty, it also laid the groundwork for later reflection on whether and how digital interventions could be reconciled with relational models of care. This ongoing tension between belief in Mamma Mia's potential and doubt about its fit with professional values is explored in the next theme.

## Theme 2: Balancing Belief and Doubt – Navigating Digital Care in Relational Professions

This theme captures a value-based tension in how midwives and PHNs positioned Mamma Mia in relation to their professional identity. Participants' accounts reflected not only assessments of the intervention itself but also deeper reflections on how digital care aligns with core professional values within MCHCs, such as relational presence, personal connection, and clinical judgment.

Several participants framed Mamma Mia as a useful supplement to existing services rather than a replacement for relational care. From this perspective, the intervention was seen as offering low-threshold, continuous support that could extend care beyond scheduled consultations and align with perinatal women's everyday digital practices. As one PHN explained: *"I imagine that Mamma Mia can be a supplement to our care at the MCHC... the app provides awareness... it supports continuity for those who use it."* (PHN, focus group 5). Others emphasized the app's potential to promote emotional awareness and self-reflection: *"It helps women recognize normal emotional fluctuations while also identifying when additional support may be needed"* (PHN, focus group 6).

At the same time, participants' endorsement was marked by ambivalence. Some questioned whether promoting a digital intervention was compatible with their identity as relational professionals, especially given their responsibility to address digital overuse and technoference in families. This created a perceived contradiction between advocating reduced screen use and simultaneously endorsing a digital intervention. This tension became more pronounced when participants felt expected to support an intervention they had not yet integrated into their clinical reasoning. One midwife reflected: *"The app's themes are fantastic, but I sometimes wonder—should we focus on strengthening midwifery-led support instead?"* (Midwife, focus group 6).

Rather than indicating resistance to innovation, such reflections suggested an effort to safeguard professional values and ensure that digital tools did not displace relational, person-centred care. These concerns were further embedded in broader organisational realities. Participants described limited capacity to respond to women with mental health needs and a lack of clear referral pathways, which constrained their confidence in recommending the intervention: *"We lack a stepped-care model, there's no clear pathway for women who struggle."* (PHN, focus group 1)

In contrast, participants with a more integrative stance emphasized how *Mamma Mia* could complement their work by maintaining contact and support between consultations, particularly for women who might otherwise hesitate to seek help: *"It can keep the door open for women between visits or for those who wouldn't otherwise seek help."* (Midwife, focus group 2). Across accounts, participants did not reject digital care outright but called for clearer role definitions, shared reflection, and professional ownership of how digital interventions should be framed, introduced, and followed up. Taken together, this theme suggests that successful adoption depended less on the intervention's content and more on whether its use could be reconciled with relational care models and professional identity.

Importantly, this tension between professional identity and innovation was significant, but not static. As participants gained experience with *Mamma Mia* and observed women's responses, many described a shift from initial scepticism toward greater confidence and cautious optimism regarding the intervention's role in everyday practice. This process of renegotiation is explored further in the next theme.

## Theme 3: Learning by Doing and Support – Future Optimism

Despite initial uncertainty, participants described how practical experience, collaboration, and sustained implementation gradually fostered and contributed to confidence in using and guiding *Mamma Mia*. Repeated exposure and the ongoing

interaction with the intervention made it easier for many to introduce the app, tailor its use to individual women's needs, and integrate it into routine consultations without compromising the relational care. As one midwife said: "*It was a bit overwhelming in the beginning, but I think it's going very smoothly now*." (Midwife, focus group 6). Another added: "*Once you find your rhythm, it becomes a valuable part of midwifery care*" (Midwife, focus group 5).

This learning-by-doing process served as a mechanism for professional reassurance, enabling practitioners to test the intervention in practice, evaluate its relevance, and gradually develop a sense of ownership and competence. Confidence was not described as emerging from training alone, but from experiencing how *Mamma Mia* worked in real encounters with women.

Structured implementation support played a decisive role in sustaining this transition. Regular supervision, access to technical assistance, user manuals, and opportunities for peer exchange were described as critical, particularly in smaller clinics, with fewer colleagues or lower birth volumes. As one PHN noted: "*If it feels overly complicated, motivation can start to fade. That's why supervision and support are so important.*" (PHN, focus group 3). Support structures appeared to function as stabilizing scaffolds, enabling professionals to persist through initial challenges rather than abandon the intervention.

With experience, *Mamma Mia* was increasingly reframed from a perceived burden to a meaningful supplement to care. Positive feedback from women using the program reinforced professional endorsement and strengthened confidence in its relational value. One midwife described how guided sessions altered the quality of consultations: "*Now that I have had quite a few of the guidance sessions in week 28 and 32 with women who are part of Mamma Mia... it affects communication quite positively. We consciously enter a conversation about the app and their experiences. That's the positive thing—and it can lead to very good conversations*." (Midwife, focus group 3).

The shift from scepticism to cautious acceptance was therefore grounded in experiential validation. Participants viewed *Mamma Mia* as filling gaps between consultations and extending support across pregnancy and early parenthood, without replacing core relational practices. Involvement in the study and implementation process was also perceived to enhance collaboration between midwives and PHNs by providing a shared intervention across care phases: "*It ensures continuity—support doesn't stop at birth but continues into early parenthood. Having something consistent throughout both stages fosters collaboration and ensures ongoing support*" (Midwife, focus group 3).

While challenges remained, particularly related to recruitment and time constraints, participants expressed a forward-looking, yet conditional optimism. Their confidence appeared closely tied to continued opportunities for learning, reflection, and support, suggesting that sustained implementation may depend less on initial enthusiasm and more on the presence of enduring organizational and professional scaffolding.

## Discussion

This study examined the experiences of midwives and public health nurses during the early implementation of the guided *Mamma Mia* intervention. The reflexive thematic analysis developed three interrelated themes that illuminate core aspects of this initial phase: *Broken Expectation – Navigating the Training-Practice Gap*, *Balancing Belief and Doubt – Navigating Digital Care in Relational Professions*, and *Learning by Doing and Support – Future Optimism.* Together, these themes illuminate tensions between training and clinical application, professional identity and digitalization, and the value of experiential learning and support.

A consistent finding was the perceived disconnect between the initial training and the realities of guiding women through Mamma Mia. While most acknowledged the value of receiving background information about Mamma Mia, they emphasized that the training was overly theoretical, too focused on implementation activities and logistics, and lacked sufficient focus on how to apply the intervention in actual consultations. For PHNs who primarily meet women postpartum and midwives with few annual births, long delays between training and first use of the intervention exacerbated this gap, leading to diminished recall and confidence. This pattern reflects the well-documented problem of limited "training

transfer", in which reduced knowledge, skills, and attitudes gained in a training context do not translate to practice [31]. From a CFIR perspective, this challenge spans the Process domain (planning and execution) and the Characteristics of Individuals domain (self-efficacy). Inadequate training, sequencing of activities, and insufficient post-training support weaken professionals' confidence in method-specific skills and can act as significant barriers to implementation [11,32]. CFIR highlights that an incremental or iterative approach can break an innovation into manageable parts for implementation, reducing perceived complexity and difficulty. Complex innovations especially benefit from such stepwise adoption, allowing practitioners time to work, learn, and build confidence through early successes [11,33].

Participants called for more experiential and context-sensitive training, including case-based discussions, role-plays, demonstrations, and peer reflection, ideally embedded in local clinical routines. These preferences align with evidence showing that training closely matched to practice realities enhances implementation success [Leeman et al., 2017; [17]. As [34] note, professionals benefit from detailed, actionable training, supported by opportunities for feedback and clarification, resonating with CFIR's reflecting and evaluating construct [11,33]. Synchronizing training with early implementation can further strengthen knowledge retention and skill application, and for guided digital interventions specifically, ongoing reflection, supervision, and feedback loops are crucial to sustain quality and confidence [8,10]. Furthermore, Leeman et al. (2017) and Jiang et al. [17] emphasize the need for training to reflect the diversity and complexity of real-world practice. This resonates with the reflecting and evaluating construct within CFIR's Process domain, which promotes shared learning and iterative adaptation [11]. Without such mechanisms, even well-designed interventions may falter in the face of real-world complexity and uncertainty. In short, training for the guided version of *Mamma Mia* should be redesigned with greater emphasis on experiential learning, post-training support, and better timing to reduce the lag between training and first use.

The implementation of Mamma Mia also sparked ambivalence, revealing a deeper tension between digital innovations and core professional values. While some participants welcomed the intervention as a timely and useful supplement to care, others worried it might erode the relational foundation of maternal and child health work. This duality is consistent with implementation guidance suggesting adoption is more likely when digital tools are perceived as complementary rather than substitutive to in-person care [15]. It also echoes broader nursing debates: digital technologies can be perceived as incompatible with traditional ideals of compassionate, hands-on care, contributing to reluctance to adopt digital approaches [35]. As relational professions grounded in embodied presence and interpersonal connection, midwifery and public health nursing may therefore view digital interventions as challenging their core ethos, human connection, care continuity, and clinical presence [15,35]. Participants described an internal conflict, recognizing Mamma Mia's potential while questioning whether promoting a digital tool aligned with their professional identity, highlighting the need for sensitive, identity-aware implementation strategies [35].

Framing Mamma Mia as a supportive tool rather than a replacement for traditional care was key to fostering acceptance. Communicating this distinction, and creating space for value-focused discussions in supervision appear to strengthen engagement and adoption [32]. The CFIR similarly emphasizes conveying an innovation's relative advantage and alignment with existing practices [11]. Importantly, guided digital innovations can provide continuity, personalization, and deepen care relationships rather than reduce guidance to task-oriented checklists [36,37]. Our participants' concerns about time, presence, and relational quality map closely onto this principle. In addition, both Agnew [36] and Booth et al. [35] advocate co-design with professionals, patients, and carers so that digital solutions augment relational care and preserve core nursing values [36,35,37]. These arguments support practical steps identified in our data: tailoring workflows locally, providing reflective supervision, and articulating clear professional roles in guided digital care.

Attitudes are a strong driver of health professionals' intention to use and recommend digital interventions [5]. To increase adoption, it is essential to communicate the value of digital tools for professionals, not only for patients or health care systems, and to continue these conversations in supervision to support reflection and shared learning [6,5]. Implementation climate, whether staff feel the innovation is expected, supported, and recognized, also shapes engagement [32,14]. Without clear leadership, room for reflection, and shared meaning-making, practitioners may feel caught between expectations to

promote a tool they did not choose and their commitment to patient-centred care. Recognizing ambivalence not as resistance, but as a legitimate expression of professional values allows for a more nuanced adoption pathway. Trust in digital interventions may grow less from persuasion than from repeated exposure, guided experience, positive responses from women, and peer validation, exactly the pattern we observed as participants began to accrue "small wins".

Peer support, manuals, and technical assistance were consistently cited as enablers. For smaller teams in particular, cross-professional collaboration between midwives and PHNs was invaluable for building confidence and sustaining engagement. Seeing women respond positively, through questions, feedback, or active use, reinforced belief in the tool and helped shift perceptions of *Mamma Mia* from a time-limited project to a sustainable part of care. Prior work shows that perceptions of digital tools often shift from barrier to facilitator once integration into workflows improves [38]. At the organizational level, participants noted the importance of supportive structures, including low-threshold access to support, supervision, and visible leadership engagement. Implementation is most successful when researchers and practitioners work closely together, fostering shared understanding and responsiveness to real-world constraints [39]. Several participants also noted Mamma Mia's potential to strengthen continuity between antenatal and postnatal care, though some emphasized the need for clearer referral pathways and a stepped-care model in their municipality. These observations align with WHO recommendations for integrating perinatal mental health within routine services through contextually adapted, collaborative approaches [1].

Finally, our findings have training, leadership, and policy implications. The urge for nurses to accelerate their transformation into a digitally enabled profession calls for education, leadership, and pragmatic implementation skills [36,35,6,37]. In the context of *Mamma Mia*, these perspectives reinforce the need to (a) time training close to first use and adapt the training to the context and profession while maintaining the collaborative element between midwives and PHNs, (b) provide structured, reflective supervision and low threshold support, (c) codesign local workflows that preserve time for relational care and implementation, and (d) articulate clear roles and referral pathways so guided digital care complements, rather than competes with, routine practice.

Taken together, these findings highlight the dynamic and evolving process of implementing digital health interventions within relational professions. The interplay between expectations, professional identity, and real-world practice reveals both the structural and emotional labour required of healthcare personnel as they navigate new roles and implementation demands. The initial mismatch between training and practice did not entirely undermine adoption, but it amplified the need for tailored support and contextual alignment. In sum, embedding guided digital interventions like *Mamma Mia* in MCHCs requires identity-aware, co-designed strategies that protect time for care, cultivate relational skills, and provide ongoing support to translate training into practice, thereby enhancing continuity without diluting the relational core of midwifery and PHN practice [35, England, 2019,15,37].

## Strengths and Limitations

A key strength of this study lies in its qualitative design and the use of reflexive thematic analysis, which allowed for a rich exploration of participants' experiences and the nuanced complexities of implementing a guided version of Mamma Mia. The study involved both midwives and PHNs, offering a diverse perspective across the perinatal continuum. Moreover, the timing of the interviews, after initial implementation, provided insight into both the anticipatory and experiential aspects of integrating Mamma Mia into clinical practice. However, the findings should be interpreted within the context of certain limitations. First, the findings reflect experiences from a specific national context, potentially limiting transferability. Using an existing framework in implementation science is considered a significant strength, enabling deeper exploration of the implementation process and capturing contextual factors that influence the implementation of blended digital interventions [32]. Second, the participants were self-selected into the study, which may have resulted in the underrepresentation of those with lower levels of readiness and capacity to implement the intervention, and possibly more negative or passive experiences. The perceptions at this specific time point may change, and future studies are needed to explore fidelity,

adaptation, and implementation over time. There is also an important role for future research in efforts to further understand the facilitators and barriers to developing strategies to promote sustainment. Finally, although the study aimed to capture a range of views, it is possible that organizational or structural influences were not fully explored due to the focus on individual experiences and the implementation of the specific intervention.

## Conclusions

This study contributes to the growing body of literature on digital health implementation by highlighting the significance of aligning training with real-world practice, addressing professional values, and supporting learning and implementation over time. The findings underscore the importance of involving practitioners before and early in the implementation process, ensuring adequate preparation, and creating a supportive environment that fosters both confidence and curiosity. While digital tools such as Mamma Mia hold great potential for enhancing continuity and access in perinatal care, their success depends on thoughtful integration into everyday clinical practice and the relational context in which care unfolds. Future efforts should focus on sustaining implementation through supervision, leadership engagement, and flexible models that honour the professional judgment of midwives and PHNs.

## Supporting information

**S1 File. Supplementary file on codes, categories, and overarching themes.**
(DOCX)

**S2 File. COREQ checklist.**
(PDF)

## Acknowledgments

We thank the health personnel for sharing their experiences, the leaders for facilitating participation, and master student Anniken Tidemann Gogstad for transcription support.

## Author contributions

**Conceptualization:** Ellen Solstad Olavesen, Silje Marie Haga, Thea Sundrehagen, Filip Drozd.

**Formal analysis:** Ellen Solstad Olavesen, Sølvi Helseth, Silje Marie Haga, Filip Drozd.

**Funding acquisition:** Silje Marie Haga, Filip Drozd.

**Investigation:** Ellen Solstad Olavesen, Thea Sundrehagen.

**Methodology:** Ellen Solstad Olavesen, Sølvi Helseth, Silje Marie Haga, Thea Sundrehagen, Filip Drozd.

**Project administration:** Silje Marie Haga, Filip Drozd.

**Supervision:** Sølvi Helseth, Silje Marie Haga, Filip Drozd.

**Validation:** Thea Sundrehagen, Filip Drozd.

**Writing – original draft:** Ellen Solstad Olavesen.

**Writing – review & editing:** Ellen Solstad Olavesen, Sølvi Helseth, Silje Marie Haga, Thea Sundrehagen, Filip Drozd.

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
