## [Decision Letter · Decision Letter 0]

9 Dec 2025

Response to Reviewers'. This file does not need to include responses to any formatting updates and technical items listed in the 'Journal Requirements' section below.'. This file does not need to include responses to any formatting updates and technical items listed in the 'Journal Requirements' section below.* A marked-up copy of your manuscript that highlights changes made to the original version. You should upload this as a separate file labeled 'Revised Manuscript with Track Changes'.'.* An unmarked version of your revised paper without tracked changes. You should upload this as a separate file labeled 'Manuscript'.'. If you would like to make changes to your financial disclosure, competing interests statement, or data availability statement, please make these updates within the submission form at the time of resubmission. Guidelines for resubmitting your figure files are available below the reviewer comments at the end of this letter. We look forward to receiving your revised manuscript. Kind regards, Mubarick Nungbaso AsumahAcademic EditorPLOS Digital Health Mubarick AsumahAcademic EditorPLOS Digital Health Leo Anthony CeliEditor-in-ChiefPLOS Digital Healthorcid.org/0000-0001-6712-6626  **Journal Requirements:** If the reviewer comments include a recommendation to cite specific previously published works, please review and evaluate these publications to determine whether they are relevant and should be cited. There is no requirement to cite these works unless the editor has indicated otherwise.  **Additional Editor Comments (if provided):****Reviewers' Comments:** Reviewer's Responses to Questions

**Comments to the Author**

1. Does this manuscript meet PLOS Digital Health’s publication criteria? Is the manuscript technically sound, and do the data support the conclusions? The manuscript must describe methodologically and ethically rigorous research with conclusions that are appropriately drawn based on the data presented.? Is the manuscript technically sound, and do the data support the conclusions? The manuscript must describe methodologically and ethically rigorous research with conclusions that are appropriately drawn based on the data presented.

Reviewer #1: Yes

Reviewer #2: Yes

Reviewer #3: Yes

2. Has the statistical analysis been performed appropriately and rigorously?

Reviewer #1: Yes

Reviewer #2: N/A

Reviewer #3: No

3. Have the authors made all data underlying the findings in their manuscript fully available (please refer to the Data Availability Statement at the start of the manuscript PDF file)?

The PLOS Data policy requires authors to make all data underlying the findings described in their manuscript fully available without restriction, with rare exception. The data should be provided as part of the manuscript or its supporting information, or deposited to a public repository. For example, in addition to summary statistics, the data points behind means, medians and variance measures should be available. If there are restrictions on publicly sharing data—e.g. participant privacy or use of data from a third party—those must be specified.requires authors to make all data underlying the findings described in their manuscript fully available without restriction, with rare exception. The data should be provided as part of the manuscript or its supporting information, or deposited to a public repository. For example, in addition to summary statistics, the data points behind means, medians and variance measures should be available. If there are restrictions on publicly sharing data—e.g. participant privacy or use of data from a third party—those must be specified.

Reviewer #1: Yes

Reviewer #2: Yes

Reviewer #3: Yes

4. Is the manuscript presented in an intelligible fashion and written in standard English?

Reviewer #1: Yes

Reviewer #2: Yes

Reviewer #3: Yes

Reviewer #1: This is a well-conducted and timely qualitative study exploring midwives’ and public health nurses’ experiences with implementing a guided digital intervention in perinatal care. The reflexive thematic analysis is rigorous, the methods transparent, and the findings provide valuable insight into the real-world challenges of translating digital health tools into relational care settings. The discussion effectively connects the themes to implementation science frameworks, though it would benefit from a sharper balance between theory and participants’ lived experiences, and a stronger focus on actionable strategies for training, supervision, and leadership support. See my attached detailed review.

Reviewer #2: Thank you for your great work.

I prefer to add some previous published researches whom used the QualCoder software in their works, and a little argument why you choose it specifically, is it for technical issues or else? For example other qualitative data analysis software offer some quireis which may be helpful for your study like Word Clouds; which is for to frequent words at the transcripts.

My suggestion is about quotes: if the data allow for choosing and using of single quote for each theme to be representative, it will be more informative, and due to reflexivity you can use the "big quote" to be rich in information related to the theme itself. Also, I suggest for more reflexive content to write about the prevalence, richness, and depth of themes; and use the result of these three parameters in your results, and discuss it at discussion section.

Sincerely

Reviewer #3: i have some minor suggestions:

1- Overly descriptive introduction: Could be shortened or restructured to emphasize the novelty and gap more directly.

2- Method clarity: While reflexivity is well described, details on how saturation was assessed and how theme validity was verified could be expanded.

3- Discussion redundancy: Some subsections repeat points about training gaps and professional identity; could be tightened.

4- Formatting: Several typographical inconsistencies (extra spacing, line breaks, repeated author initials in the rigour section).

5- Conclusion: Strong but could include explicit policy or training recommendations for wider implementation beyond Norway.

**Do you want your identity to be public for this peer review?** For information about this choice, including consent withdrawal, please see our Privacy Policy..

Reviewer #1: **Yes:**Victor Ifechukwude AgboliVictor Ifechukwude AgboliVictor Ifechukwude AgboliVictor Ifechukwude Agboli

Reviewer #2: **Yes:**Jehad Omar AbualrobJehad Omar AbualrobJehad Omar AbualrobJehad Omar Abualrob

Reviewer #3: No

**Figure resubmission:**  While revising your submission, we strongly recommend that you use PLOS’s NAAS tool (https://ngplosjournals.pagemajik.ai/artanalysis) to test your figure files. NAAS can convert your figure files to the TIFF file type and meet basic requirements (such as print size, resolution), or provide you with a report on issues that do not meet our requirements and that NAAS cannot fix. 

**Reproducibility:** To enhance the reproducibility of your results, we recommend that authors of applicable studies deposit laboratory protocols in protocols.io, where a protocol can be assigned its own identifier (DOI) such that it can be cited independently in the future. Additionally, PLOS ONE offers an option to publish peer-reviewed clinical study protocols. Read more information on sharing protocols at https://plos.org/protocols?utm_medium=editorial-email&utm_source=authorletters&utm_campaign=protocols To enhance the reproducibility of your results, we recommend that authors of applicable studies deposit laboratory protocols in protocols.io, where a protocol can be assigned its own identifier (DOI) such that it can be cited independently in the future. Additionally, PLOS ONE offers an option to publish peer-reviewed clinical study protocols. Read more information on sharing protocols at https://plos.org/protocols?utm_medium=editorial-email&utm_source=authorletters&utm_campaign=protocols

---

## [Decision Letter · Decision Letter 1]

19 Mar 2026

Midwives’ and Public Health Nurses’ Experiences of Implementing a Guided Version of a Digital Intervention, Mamma Mia, in Maternity and Child Health Care Services: A Reflexive Thematic Analysis

PDIG-D-25-00667R1

Dear MSc Olavesen,

We are pleased to inform you that your manuscript 'Midwives’ and Public Health Nurses’ Experiences of Implementing a Guided Version of a Digital Intervention, Mamma Mia, in Maternity and Child Health Care Services: A Reflexive Thematic Analysis' has been provisionally accepted for publication in PLOS Digital Health.

Best regards,

Mubarick Nungbaso Asumah

Academic Editor

PLOS Digital Health

**Additional Editor Comments (if provided):**

**Reviewer Comments (if any, and for reference):**

Reviewer's Responses to Questions

**Comments to the Author**

Reviewer #1: All comments have been addressed

Reviewer #2: All comments have been addressed

publication criteria? Is the manuscript technically sound, and do the data support the conclusions? The manuscript must describe methodologically and ethically rigorous research with conclusions that are appropriately drawn based on the data presented.? Is the manuscript technically sound, and do the data support the conclusions? The manuscript must describe methodologically and ethically rigorous research with conclusions that are appropriately drawn based on the data presented.

Reviewer #1: Yes

Reviewer #2: Yes

3. Has the statistical analysis been performed appropriately and rigorously?

Reviewer #1: Yes

Reviewer #2: N/A

4. Have the authors made all data underlying the findings in their manuscript fully available (please refer to the Data Availability Statement at the start of the manuscript PDF file)?

The PLOS Data policy requires authors to make all data underlying the findings described in their manuscript fully available without restriction, with rare exception. The data should be provided as part of the manuscript or its supporting information, or deposited to a public repository. For example, in addition to summary statistics, the data points behind means, medians and variance measures should be available. If there are restrictions on publicly sharing data—e.g. participant privacy or use of data from a third party—those must be specified.requires authors to make all data underlying the findings described in their manuscript fully available without restriction, with rare exception. The data should be provided as part of the manuscript or its supporting information, or deposited to a public repository. For example, in addition to summary statistics, the data points behind means, medians and variance measures should be available. If there are restrictions on publicly sharing data—e.g. participant privacy or use of data from a third party—those must be specified.

Reviewer #1: Yes

Reviewer #2: Yes

5. Is the manuscript presented in an intelligible fashion and written in standard English?

Reviewer #1: Yes

Reviewer #2: Yes

Reviewer #1: The authors have addressed the main concerns from my first review, and the revised manuscript is now clear, coherent, and methodologically sound. I have no further comments.

Reviewer #2: Thank you for your response. Best wishes

**Do you want your identity to be public for this peer review?** For information about this choice, including consent withdrawal, please see our Privacy Policy..

Reviewer #1: **Yes:**Victor Ifechukwude AgboliVictor Ifechukwude AgboliVictor Ifechukwude AgboliVictor Ifechukwude Agboli

Reviewer #2: **Yes:**Jehad Omar AbualrobJehad Omar AbualrobJehad Omar AbualrobJehad Omar Abualrob
